# RETRIEVAL-GUIDED CROSS-VIEW IMAGE SYNTHESIS

## ABSTRACT

Cross-view image synthesis involves generating new images of a scene from different viewpoints or perspectives, given one input image from other viewpoints. Despite recent advancements, there are several limitations in existing methods: 1) reliance on additional data such as semantic segmentation maps or preprocessing modules to bridge the domain gap; 2) insufficient focus on view-specific semantics, leading to compromised image quality and realism; and 3) a lack of diverse datasets representing complex urban environments. To tackle these challenges, we propose: 1) a novel retrieval-guided framework that employs a retrieval network as an embedder to address the domain gap; 2) an innovative generator that enhances semantic consistency and diversity specific to the target view to improve image quality and realism; and 3) a new dataset, VIGOR-GEN, providing diverse cross-view image pairs in urban settings to enrich dataset diversity. Extensive experiments on well-known CVUSA, CVACT, and new VIGOR-GEN datasets demonstrate that our method generates images of superior realism, significantly outperforming current leading approaches, particularly in SSIM and FID evaluations.

## 1 INTRODUCTION

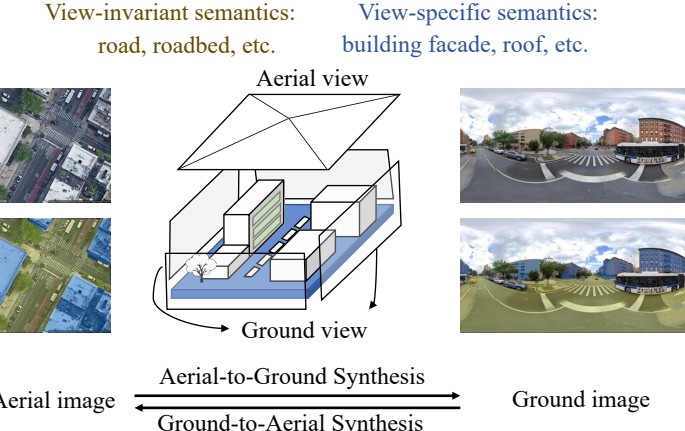

Figure 1: Cross-view image synthesis: Illustrating view-invariant semantics and view-specific semantics in aerial or ground view .

Cross-view image synthesis aims to generate images from a new perspective or viewpoint that differs from the original image, which synthesizes images from a given view (e.g., aerial or bird's eye view) to a target view (e.g., street or ground view), even when the target viewpoint was not originally captured. It offers a wide range of applications, such as autonomous driving, robot navigation, 3D reconstruction Mahmud et al. (2020), virtual/augmented reality Bischke et al. (2016), urban planning Máttyus et al. (2017), etc. In this paper, we probe into the ground-to-aerial / aerial-to-ground view synthesis based on a given source-view image (as illustrated in the upper half of Figure 1). This task presents significant challenges, as it requires the model to comprehend and interpret the scene's geometry and object appearances from one view, and then reconstruct or generate a realistic image from a different viewpoint.

While promising, several key challenges plague existing cross-view image synthesis methods. 1) Reliance on additional data. Existing methods often rely on extra information like semantic segmentation maps Regmi & Borji (2018); Tang et al. (2019); Wu et al. (2022) or preprocessing modules like polar-transformation Lu et al. (2020); Toker et al. (2021); Shi et al. (2022) to bridge the domain gap between different views. These extra steps not only increase the computational burden but also complicate the reverse generation process (e.g., ground-to-aerial synthesis). 2) Limited focus on view-specific semantics. Most models primarily focus on view-invariant semantics between views, neglecting the importance of view-specific semantics. View-invariant semantics refer to elements that maintain fundamental similarity across views despite visual differences, such as roads viewed from aerial and ground views (highlighted in translucent yellowish-green in Figure 1's lower half). Conversely, view-specific semantics represent objects with drastically different appearances across viewpoints, exemplified by a building's roof in aerial view versus its facade in ground view (highlighted in translucent blue in Figure 1's lower half). While view-specific semantics help establish correspondence between views, the neglect of view-specific semantics limits the fidelity and realism of the synthesized images. 3) Lack of diverse datasets. Existing datasets for cross-view image synthesis primarily focus on rural and suburban areas, overlooking the complexities of urban environments. This lack of diversity in training data makes it challenging to develop models that can effectively synthesize images in more realistic and challenging scenarios.

In this study, we propose a new cross-view image synthesis method that does not require semantic segmentation maps or preprocessing modules while generating high-fidelity, realistic target-view images by fully leveraging view-invariant and view-specific semantics. Inspired by the retrieval task's nature of measuring similarity in view-invariant semantics, we introduce a retrieval network as an embedder to encode these semantics and guide the generation process. This approach obviates the need for preprocessing or segmentation maps for cross-view image pairs. To enhance image quality and realism, our method incorporates view-specific semantics, by adopting noise and modulated style to diversify visual features. We fuse retrieval embedding and style at various layers to improve consistency and image quality. Additionally, to address the scarcity of urban datasets for cross-view image synthesis, we introduce VIGOR-GEN, a derived urban dataset. We validate our proposed method through comprehensive experiments on CVUSA Zhai et al. (2017), CVACT Liu & Li (2019), and the more challenging VIGOR-GEN dataset. Our model generates more realistic images and significantly outperforms state-of-the-art methods, particularly in terms of SSIM and FID. Extensive ablation studies corroborate the efficacy of each component in our method.

The main contributions are summarized as follows:

- Retrieval-Guided Framework for Bridging Domain Gap. We introduce a retrieval-guided framework that leverages a retrieval network as an embedder. This network is trained to measure the similarity of view-invariant between different views, effectively bridging the domain gap without needing semantic segmentation maps or preprocessing modules. Our model simplifies the synthesis process and makes reverse generation (e.g., ground-to-aerial) more straightforward.

- Novel Generator for Enhanced Semantic Consistency and Diversity. Our proposed method includes a new generator that incorporates both retrieval embedding and style information at various layers. This approach improves the correspondence between views by leveraging view-invariant semantics captured by the retrieval network, while also enhancing the diversity and realism of view-specific semantics using noise and modulated style techniques. This leads to synthesized images with higher fidelity and a more natural appearance.

- New Dataset for Urban Environments (VIGOR-GEN). We build a new derived dataset called VIGOR-GEN, which provides a more challenging and realistic setting for training and evaluating cross-view image synthesis models, pushing the boundaries of the field beyond existing rural and suburban datasets. Our method demonstrates superior performance in synthesizing photo-realistic images from a single input image in another view, as evidenced by its performance on well-known datasets, and the new VIGOR-GEN datasets.

## 2 RELATED WORK

**Semantic-guided Cross-view Synthesis** The first pipeline is to apply the semantic segmentation maps of the target-view images to guide the generative model. Zhai Zhai et al. (2017) proposed a

linear transformation module to generate a panorama via supervised information from a transformed semantic layout of aerial images. Regmi and Borji Regmi & Borji (2018) designed two cGAN models, X-fork and X-seq, for simultaneously predicting the target image as well as the semantic map. Tang Tang et al. (2019) regarded cross-view image synthesis as an image-to-image translation task. This work applied the semantic map of the target view and the source view image as inputs and then obtained the predicted target images. To generate 360-degree panorama images, Wu Wu et al. (2022) proposed PanoGAN as well as a new discrimination mechanism. Zhu Zhu et al. (2023) proposed a Parallel Progressive GAN to stabilize the training of cross-view image synthesis and thus generated rich details.

**Preprocessing-Guided Cross-view Synthesis**   Another pipeline involves a preprocessing module to assimilate the source view image into the target view image. Lu Lu et al. (2020) proposed a projection transformation module that is trained by height and semantic information estimated from aerial images. However, this approach requires ground-truth height supervision for the dataset and carries a complicated pipeline. Toker Toker et al. (2021) first applied the polar transformation proposed by Shi Shi et al. (2019) to cross-view image synthesis, which greatly reduces the domain gap between two views. Besides, Toker Toker et al. (2021) proposed a new multi-tasks framework Coming-Down-to-Earth (CDE) for synthesis, where they postulated that retrieval and synthesis tasks are orthogonal. This approach further improves the correspondence of generation but fails to produce better image detail and quality. Shi Shi et al. (2022) proposed an end-to-end network that employs a learnable geographic projection module to learn the projection relationship from the aerial view to the ground view, and then feed the manipulated image into the later generator.

As a striking difference from existing works, without the help of semantic maps and preprocessing, our model can synthesize a more realistic target-view image and retain rich details, capable of realizing the mutual generation of ground panorama and aerial image.

**Generative Model**   In recent years, diffusion model Rombach et al. (2022); Croitoru et al. (2023); Ramesh et al. (2022); Saharia et al. (2022) achieved great success, which produces higher quality images at the cost of a large amount of resources. In addition, there are still neglected problems in cross-view image synthesis, as described in the next section. Moreover, earlier work on cross-view generation does not yield better performance with more artifacts. Therefore, it is essential to study a competitive GAN model before moving fully towards the diffusion model.

## 3 METHODOLOGY

In this section, we first introduce our architecture for cross-view image synthesis. Then, we give an overview of the proposed network in Figure 2.

### 3.1 OVERVIEW OF RETRIEVAL-GUIDED FRAMEWORK

We propose a novel cross-view image synthesis framework that leverages a pre-trained and fixed retrieval model to identify view-invariant semantics within a specific view, enabling an end-to-end program without requiring preprocessing or additional input.

The embedder, trained through contrastive learning, maps view-invariant semantics into a continuous space, allowing for fusion in the deeper layers. This approach aims to extract embeddings that minimize visual differences, ensuring a smooth transformation of view-invariant semantics from the source domain to the target domain via the generator, thereby preserving the image structure.

Moreover, the embedding can also serve as the condition in the discriminator to guide the generator to improve correspondence.

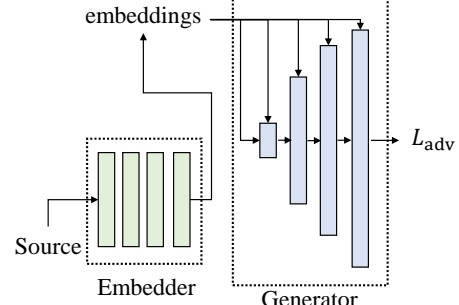

Figure 2: Overview of the proposed framework.

Meanwhile, we consider the ability of the model to generate view-specific semantics in the target domain by offering modulated style information. Although it is difficult to generate identical target-view images, our goal is to ensure that the view-invariant semantics in the generated images are consistent between the two views while the view-specific semantics remain as visually reasonable as possible.

## 3.2 NETWORK ARCHITECTURE

The overall architecture of our network is illustrated in Figure 3. It consists of two components: the mapping network and the retrieval network. **The Mapping Network**: Our network has a mapping network which has already been shown in several works Karras et al. (2019; 2020b;a); Choi et al. (2020). The mapping network learns how to transform the noise sampled from a Gaussian distribution to a new style distribution to better generate exclusive representations, thus yielding detail-enriched images. The mapping network consists of four fully connected layers with non-linearity. **The Retrieval Network**: We adopt the retrieval network proposed in Zhu et al. (2023) because of its simplicity and effectiveness. It owns stacked attention layers for better feature extraction and encoding for retrieval. We utilize its shallower version SAIG-S here. This retrieval network can settle visual differences and directly embed images from different views into a smooth space. Please refer to the original paper and the Appendix of this paper for more details.

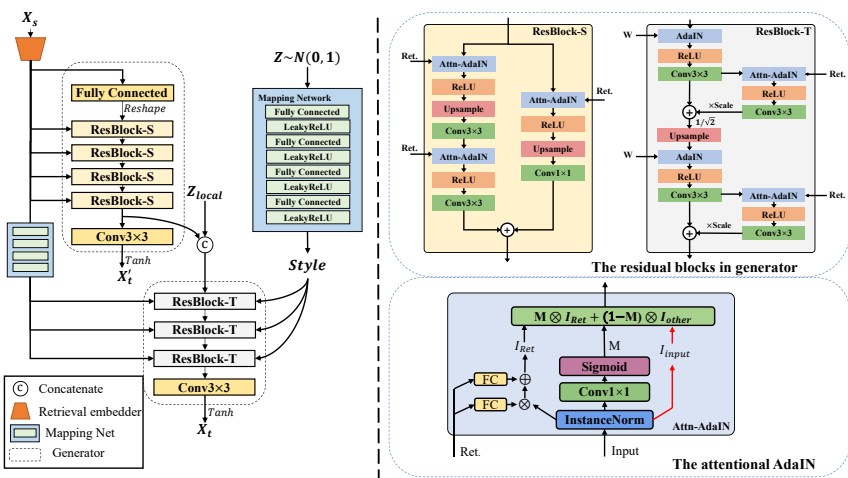

Figure 3: **Illustration of our network architecture. left**: our network consists of a structure generator, a facade generator, a mapping network, and a retrieval embedder. **right-top**: the residual blocks in our generator. **right-bottom**: the attentional AdaIN in different residual blocks.

## 3.3 STRUCTURE & FACADE GENERATION

**Two-stage generation**  In general, the generative model controls the generation of structures at low resolutions ($\leq 32 \times 32$), while features such as facade and color will be affected in higher resolutions ($\geq 32 \times 32$) Karras et al. (2020b); Richardson et al. (2021); Yang et al. (2022). Therefore, we refine the goals of the generator: at low resolution, the generator focuses on projecting the view-invariant semantics into target-view space. Once the approximate structure of the target view has been generated, the generator then turns its attention to how to generate facades while preserving identity.

**Attentional AdaIN**  The embedding extracted by the retrieval model contains the semantic information of the location. Some work Huang & Belongie (2017); De Vries et al. (2017); Tao et al. (2022); Park et al. (2019); Zhu et al. (2020) has explored how to incorporate the latent code into feature maps to acquire target images. To better inject identity information into the image, we perform some changes to AdaIN Huang & Belongie (2017) to make feature maps more semantically consistent with the given source image. Given an input $\mathbb{X} \in \mathbb{R}^{n \times c \times h \times w}$, we first normalize it into zero mean and unit deviation:

$$\hat{\mathbb{X}} = \frac{\mathbb{X} - \mu_{nc}}{\sigma_{nc}}, \mu_{nc} = \frac{1}{hw}\sum_{hw}\mathbb{X}, \sigma_{nc} = \sqrt{\frac{1}{hw}\sum_{hw}(\mathbb{X} - \mu_{nc}^2) + \epsilon} \tag{1}$$

where $\epsilon$ is a small constant to prevent the divisor from being zero, $\mu_{nc}$ denotes the mean and $\sigma_{nc}$ denotes the variance.

Subsequently, the modulation parameters $\gamma$ and $\beta$ are learned by MLP from the retrieval feature $\hat{r}$ :

$$\gamma_r = MLP_\gamma(\tilde{r}), \beta_r = MLP_\beta(\tilde{r}) \tag{2}$$

Then, the denormalization can be realized as follows:

$$\hat{\mathbb{X}}_r = \gamma_r\hat{\mathbb{X}} + \beta_r \tag{3}$$

To decide which region and to what extent it can reinforce the retrieval embedding on the image feature, we utilize input $\mathbb{X}$ to learn to obtain a weight map $M$. It can be described as:

$$M = Sigmoid(Conv(\hat{\mathbb{X}})) \tag{4}$$

where $Sigmoid$ denotes the sigmoid activate function. In the ideal case, we expect the modulation of retrieval embeddings to work on the areas where the source view is relevant to the target view.

Finally, the feature maps are summed by $M$ on the pixel-wise level:

$$\tilde{\mathbb{X}} = \hat{\mathbb{X}}_r \cdot M + \hat{\mathbb{X}} \cdot (1 - M) \tag{5}$$

**Different residual modules**   Residual structures have been widely applied in prior work Regmi & Borji (2018); Tang et al. (2019); Wu et al. (2022); Shi et al. (2022); Zhu et al. (2023) on cross-view image synthesis to aid in structure generation. However, other work Karras et al. (2019; 2020b) argues that residual structures introduce varying degrees of artifacts and blurring in generation, especially in facade generation. Therefore, the modules for generating structures and facades have to be carefully considered, according to different task objectives.

For structure generation, we use a residual structure similar to previous methods, except for the use of an improved AdaIN in the normalization layer. Both the principal and residual paths are injected with retrieval embedding to facilitate the construction of the structure. For facade generation, we follow the network design of previous work Karras et al. (2019; 2020b), but the residual structure is also used. The input latent is first fed into AdaIN and the convolution layer to fuse the modulated style. The residual structure is designed to be set after the convolution layer and continue to fuse the embedding through an improved AdaIN. The residual path is then multiplied by a layer scale Touvron et al. (2021); Sauer et al. (2023) to perform gradual fading.

**Generator**   As shown in Figure 3, our generator first gains the retrieval embedding from the source images $\mathbb{X}_s$ as the input, which is then integrated into a fully connected layer and is reshaped to be equally proportional to the target image $\mathbb{X}_t$ in length and width. The latent feature synthesized by the structure generator is then concatenated with a noise vector sampled from the Gaussian distribution. The generator gradually increases the scale of the feature map and eventually converts it into an image. Each residual block in the decoder contains 1) Normalization layers integrating style information or retrieval information; 2) Convolutional layers with spectral normalization Miyato et al. (2018) and 3) Activate function.

**Discriminator**   To guide our generator to synthesize more realistic and semantically consistent images with the source image, we adopt the idea of a one-way discriminator proposed in Tao et al. (2022). It first extracts the features of the synthesized image and then concatenates them with the spatially extended embedding vector. The discriminator should assign the realistic and matching images with high scores, and the fake or mismatched images with low scores. The details of the discriminator are presented in the Appendix.

### 3.4 LOSS FUNCTION

**Discriminator Loss** Since the one-way discriminator is employed, we apply the same adversarial loss Tao et al. (2022) except for the gradient penalty to train our network.

$$
\begin{aligned}
\mathcal{L}_{adv}^{D} = & -\mathbb{E}_{\mathbb{X}\sim\mathbb{P}_r}[\min(0, -1 + D(\mathbb{X}, \tilde{\mathbf{A}}))] \\
& - (1/2)\mathbb{E}_{\hat{\mathbb{X}}\sim\mathbb{P}_g}[\min(0, -1 - D(\hat{\mathbb{X}}, \tilde{\mathbf{A}}))] \\
& - (1/2)\mathbb{E}_{\mathbb{X}\sim\mathbb{P}_{mis}}[\min(0, -1 - D(\mathbb{X}, \tilde{\mathbf{A}}))]
\end{aligned}
\tag{6}
$$

where $\tilde{\mathbf{A}}$ refers to the retrieval embeddings of the real image $\mathbb{X}$ and $\hat{\mathbb{X}}$ denotes the synthesized image.

**Generator Loss** The reconstruction loss is employed to ensure that the target $\mathbb{X}_t$ is equivalent to the final result $\mathbb{X}_r$ on a pixel-wise level. It can defined as follows:

$$
\mathcal{L}_{rec} = \|\mathbb{X}_t - \mathbb{X}_r\|^1
\tag{7}
$$

To further improve the realism, we follow the Learned Perceptual Image Patch Similarity (LPIPS) Zhang et al. (2018) loss. Thus, the perceptual loss is defined as:

$$
\mathcal{L}_{perc} = \|\phi(\mathbb{X}_t) - \phi(\mathbb{X}_r)\|^1
\tag{8}
$$

where $\phi$ denotes the pre-trained VGG network.

To ensure that the synthesized image has the same shared information as the target image, we use identity loss, which is defined as:

$$
\begin{aligned}
\mathcal{L}_{id} = & 1 - \cos(R(\mathbb{X}_r), R(\mathbb{X}_t)) \\
& + 1 - \cos(R(\mathbb{X}_r^{'}), R(\mathbb{X}_t^{'}))
\end{aligned}
\tag{9}
$$

where $cos(.,.)$ denotes the cosine similarity between the output embedding vectors and $\mathbb{X}_r^{'}$ means the low-resolution generated image. $R$ denotes the pre-trained retrieval network as in Sec. 3.2.

To prevent the model from generating repetitive content, we apply a diversity loss Mao et al. (2019); Lee et al. (2020) between a pair of local code $Z_{local}$. The diversity loss is defined as:

$$
\mathcal{L}_{div} = \frac{d_z(z_{local_1}, z_{local_2})}{d_I(G(w, z_{local_1}, \tilde{\mathbf{A}}), G(w, z_{local_2}, \tilde{\mathbf{A}}))}
\tag{10}
$$

where $d_z(.,.)$ and $d_I(.,.)$ denote the $L1$ distance between the latent codes or images, $G$ is the generator.

The adversarial loss of the generator is as follows:

$$
\mathcal{L}_{adv}^{G} = \mathbb{E}_{\hat{x}\sim\mathbb{P}_g}[D(\hat{\mathbb{X}}, \tilde{\mathbf{A}})]
\tag{11}
$$

The total loss for the generator is a weighted sum of the above losses, formulated as:

$$
\mathcal{L}_G = \mathcal{L}_{adv}^{G} + \lambda_{rec}\mathcal{L}_{rec} + \lambda_{perc}\mathcal{L}_{perc} + \lambda_{id}\mathcal{L}_{id} + \lambda_{div}\mathcal{L}_{div}
\tag{12}
$$

## 4 VIGOR-GEN DATASET

For cross-view image synthesis, the commonly used CVUSA Zhai et al. (2017) and CVACT Liu & Li (2019) datasets are primarily field and sub-urban images with an open field of view and less occlusion. The buildings on both datasets are mostly cottages or bungalows, with simple facade information. In contrast to the above datasets, the images with soaring skyscrapers in urban areas often have narrower views and more occlusions, while the complex street surroundings and building facades raise greater challenges to generative networks. To fit realistic scenarios, the cross-view image synthesis generates the need for an urban area dataset. To this end, we have collected a derived dataset of cross-view urban images, VIGOR-GEN, consisting of 103,516 image pairs.

All images are collected from Google Map API map. The dataset is mainly extended on cross-view image retrieval dataset VIGOR Zhu et al. (2021). To ensure that images synthesized across different views have the same identity as the source image, this task usually requires center-aligned image pairs to avoid ambiguities, so the original VIGOR urban dataset (which is set to be non-centrally aligned) cannot be directly applied to this task. To extend the application of this dataset, we present a derived dataset in this work so that it can be used for cross-view image synthesis. Table 1 shows the comparison of different datasets.

Table 1: The comparison of VIGOR-GEN and other existing open panorama-aerial cross-view image datasets

| Dataset | CVUSA | CVACT | VIGOR | VIGOR-GEN |
|---|---|---|---|---|
| Area | field | suburban | urban | urban |
| Satellite resolution | $750 \times 750$ | $1200 \times 1200$ | $640 \times 640$ | $640 \times 640$ |
| Panorama resolution | $1232 \times 224$ | $1664 \times 832$ | $2048 \times 1024$ | $2048 \times 1024$ |
| Roughly centered | Yes | Yes | No | Yes |
| Application | Retrieval, Generation | Retrieval, Generation | Retrieval | Retrieval, Generation |
| #Satellite Image | 44,416 | 44,416 | 90,618 | 103,516 |
| #Panorama Image | 44,416 | 44,416 | 105,214 | 103,516 |

## 5 EXPERIMENT

### 5.1 IMPLEMENTATION DETAILS

**Datesets.** We perform our experiments on the panorama-aerial dataset CVUSA, CVACT, and our newly proposed VIGOR-GEN. Following Toker et al. (2021); Shi et al. (2022), the CVUSA and CVACT consist of 44,416 image pairs with the train/test split of 35,532/8,884. The VIGOR-GEN dataset consists of 51,366 images for training and 51,250 images for testing. The resolution of the panorama is set at $128 \times 512$ in CVUSA and $256 \times 512$ in both CVACT and VIGOR-GEN. All aerial images are set to a resolution of $256 \times 256$.

**Metrics.** Following previous work Regmi & Borji (2018); Lu et al. (2020); Toker et al. (2021); Shi et al. (2022), we adopt the widely used *Structural-Similarity (SSIM)*, *Peak Signal-to-Noise Ratio (PSNR)* and *Learned Perceptual Image Patch Similarity (LPIPS)* Zhang et al. (2018) to measure the similarity at the pixel-wise level and feature-wise level, respectively. Meanwhile, the realism of the images is measured by *Fréchet Inception Distance (FID)* Heusel et al. (2017). We report the Recall@1 (R@1) in our experiment using another cross-view image retrieval model SAIG-D Zhu et al. (2023), which indicates whether the resulting images describe the same location.

**Training Details.** The experiments are implemented using PyTorch. We train our model with 200 epochs using Adam Kingma & Ba (2014) optimizer and $\beta_1 = 0.5, \beta_2 = 0.999$. The learning rate of the generator and discriminator is set to 0.0001 and 0.0004, respectively. Please refer to the Appendix for more details about training.

### 5.2 COMPARISONS WITH STATE-OF-THE-ART METHODS

We compared our method with Pix2Pix Isola et al. (2017), XFork Regmi & Borji (2018), SelectionGAN Tang et al. (2019), PanoGAN Wu et al. (2022), CDE Toker et al. (2021) and S2SP Shi et al. (2022), PPGAN Zhu et al. (2023), Sat2Density Qian et al. (2023), ControlNet Zhang et al. (2023), Instruct pix2pix Brooks et al. (2023), CrossViewDiff Croitoru et al. (2023) on CVUSA and CVACT datasets. The results are shown in Table 3 and 2. For S2SP Shi et al. (2022), it applies the geometry project equation to calculate the projection from satellite image to street-view panorama, whose inverse process is not given in the original paper, so this method will not be compared at g2a generation.

Table 3: The comparison of existing competitive methods on CVUSA and CVACT. The comparison of existing competitive methods on CVUSA and CVACT. Note that for the FoV-only model, we follow Tang et al. (2019) and obtain the final panorama, which consists of four street images with a FoV of 90 degrees. For a fair comparison, we discard the semantic maps as an input in SelectionGAN.

| Direction | Method | CVUSA | | | | | CVACT | | | | |
|---|---|---|---|---|---|---|---|---|---|---|---|
| | | SSIM↑ | PSNR↑ | LPIPS↓ | FID↓ | R@1↑ | SSIM↑ | PSNR↑ | LPIPS↓ | FID↓ | R@1↑ |
| a2g | Pix2Pix | 0.2849 | 12.14 | 0.5712 | 82.84 | 0.01 | 0.3634 | 13.37 | 0.4943 | 86.21 | 0.00 |
| | XFork | 0.3408 | 13.25 | 0.5611 | 79.75 | 6.41 | 0.3701 | 14.17 | 0.4919 | 47.98 | 8.72 |
| | SelectionGAN | 0.3278 | 13.37 | 0.5331 | 90.72 | 4.58 | 0.4705 | 14.31 | 0.5141 | 95.67 | 6.67 |
| | PanoGAN | 0.3024 | 13.67 | 0.4684 | 75.24 | 33.11 | 0.4631 | 14.18 | 0.4762 | 82.65 | 28.71 |
| | CDE | 0.2980 | 13.87 | 0.4752 | 20.63 | 85.04 | 0.4506 | 13.98 | 0.4927 | 43.96 | 65.04 |
| | S2SP | 0.3437 | 13.32 | 0.4688 | 44.15 | 10.09 | 0.4521 | 14.14 | 0.4718 | 39.64 | 29.39 |
| | PPGAN | 0.3516 | 13.91 | - | - | - | - | - | - | - | - |
| | Sat2Density | 0.3390 | 14.23 | - | 41.43 | - | 0.3870 | 14.27 | - | 47.09 | - |
| | ControlNet | 0.2770 | 11.18 | - | 44.63 | - | 0.3400 | 12.15 | - | 47.15 | - |
| | Instruct pix2pix | 0.2550 | 10.66 | - | 68.75 | - | 0.3920 | 13.12 | - | 57.74 | - |
| | CrossViewDiff | **0.3710** | 12.00 | - | 23.67 | - | 0.4120 | 12.41 | - | 41.94 | - |
| | Ours | 0.3706 | **14.33** | **0.4302** | **13.57** | **96.25** | **0.4945** | **14.55** | **0.4540** | **21.83** | **87.90** |
| g2a | Pix2Pix | 0.1956 | 15.07 | 0.6220 | 121.95 | 7.85 | 0.0870 | 14.24 | 0.6612 | 133.39 | 13.06 |
| | CDE | 0.2167 | 15.19 | 0.5706 | 121.98 | 14.73 | 0.0906 | 14.59 | 0.6689 | 160.81 | 14.99 |
| | Ours | **0.2461** | **15.77** | **0.5181** | **41.65** | **95.14** | **0.1966** | **16.29** | **0.5551** | **36.54** | **87.81** |

**Quantitative Results** For aerial-to-ground image synthesis, our method outperforms the existing methods S2SP Shi et al. (2022) by 6 points in terms of SSIM on the CVUSA dataset. For PSNR and LPIPS, our method achieves 1.01 and 0.0386 improvement, respectively. Our method outperforms the existing state-of-the-art methods CrossViewDiff Croitoru et al. (2023) by 2.33 points in terms of PSNR on the CVUSA dataset

Table 2: The comparison of existing competitive methods on our newly proposed VIGOR-GEN.

| Direction | Method | VIGOR-GEN | | | | |
|---|---|---|---|---|---|---|
| | | SSIM↑ | PSNR↑ | LPIPS↓ | FID↓ | R@1↑ |
| a2g | Pix2Pix | 0.3566 | 12.18 | 0.6114 | 100.25 | 0.01 |
| | SelectionGAN | 0.3986 | 13.16 | 0.5234 | 104.22 | 7.41 |
| | PanoGAN | 0.4031 | 13.83 | 0.5467 | 75.76 | 8.49 |
| | CDE | 0.3672 | 12.72 | 0.6108 | 78.26 | 0.22 |
| | S2SP | 0.4041 | 13.73 | 0.5422 | 69.28 | 4.54 |
| | Ours | **0.4243** | **13.91** | **0.4548** | **13.64** | **37.94** |
| g2a | Pix2Pix | 0.1885 | 13.31 | 0.5876 | 96.26 | 2.41 |
| | CDE | 0.1830 | 12.89 | 0.5734 | 95.13 | 3.25 |
| | Ours | **0.1901** | **13.99** | **0.5278** | **30.93** | **34.58** |

and 2.14 points in terms of PSNR on the CVACT dataset and gains an important improvement by 0.0825 points in terms of SSIM on the CVACT dataset. In g2a image synthesis, compared to the most competitive method CDE Toker et al. (2021), our model gains a significant improvement in LPIPS (0.5181 versus 0.5706 on CVUSA), which proves that generated images are more consistent with human visual perception. This is attributed to the embedding can be used as the condition on the discriminator to guide the generator to improve the correspondence, which does not apply to CDE Toker et al. (2021) as it introduces labeling uncertainty.

It is worth noting that our method has a larger improvement in FID compared to other models. For example, our method gains a 7.06 point improvement compared to CDE Toker et al. (2021). This is because we consider not only the view-invariant semantics across views but also the view-specific semantics of the target view, which makes the synthesized images more realistic. Especially, in ground-to-aerial image synthesis, it is challenging for other models to generate the obscured parts, resulting in a decrease in realism. A lower FID can be observed on CVUSA (41.65 versus 121.95).

Experiments are also conducted on our newly proposed urban dataset VIGOR-GEN which is more challenging due to its complex facades and inevitable occlusions. As a result, the exclusive information in one view is more complicated in an urban setting. As shown in Table 2, our method outperforms other methods in all metrics. For example, our proposed method sets the new state-of-the-art FID of 13.64 at a2g and 30.93 at g2a on VIGOR-GEN while other methods have a higher FID. For the R@1 metric, the multi-task framework CDE, which performs well on CVUSA and CVACT, almost fails in VIGOR-GEN. In other words, CDE does not fit well in urban areas while our method still produces images with higher quality.

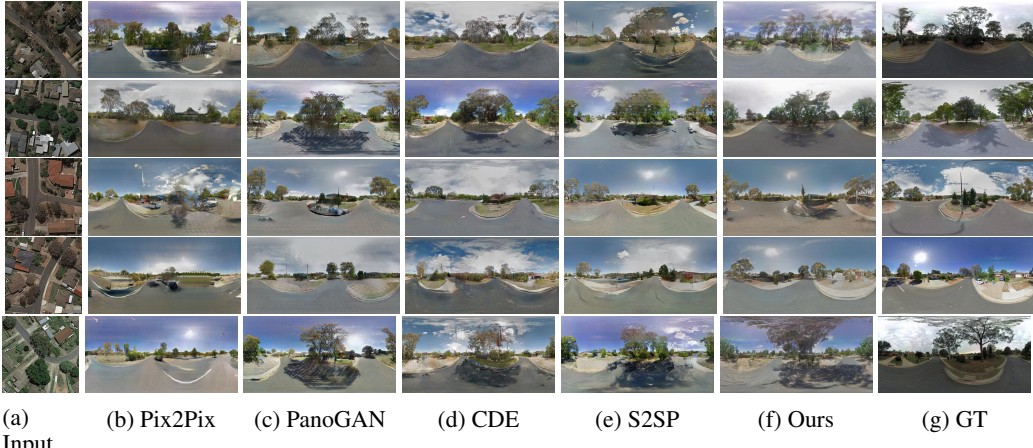

(a) Input     (b) Pix2Pix     (c) PanoGAN     (d) CDE     (e) S2SP     (f) Ours     (g) GT

Figure 4: Comparison with current methods at a2g direction on CVACT.

**Qualitative Results**    We provide the qualitative results of our method on different datasets to validate its effectiveness. From the qualitative comparison shown in Figure 4, we can observe our method generates more realistic and detailed images with fewer artifacts compared to other methods.

Compared to other methods, as shown in the first group of Figure 4, our approach generates consistent and clear roads with fewer artifacts on CVACT. This indicates that our method is capable of overcoming the visual difference. In addition, our method exhibits exceptional performance in complex scenes. For instance, in the first row in the second group of Figure 4, our method synthesizes more realistic building facades, including intricate details such as windows and doors. Other methods, by contrast, fail to produce these distinctive features in the panoramic view. The ability of our method to generate exclusive information in the target view is a result of its consideration of not only the correspondence between the source and target views but also the content difference between them. As opposed to other models that struggle to address the difference of exclusive information, our model is equally well-suited for urban areas.

Further evidence supporting our idea is that when generating aerial-view images, other methods only produce blurred border regions. As demonstrated in Figure 5, Pix2Pix Isola et al. (2017) and CDE Toker et al. (2021) generate central areas that are barely clear while introducing artifacts and blurs in the roof or non-central regions, where exclusive aerial image information resides. For more results, please refer to the Appendix.

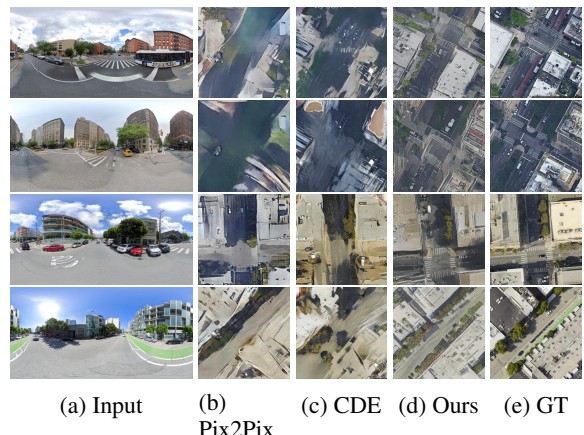

(a) Input    (b) Pix2Pix    (c) CDE   (d) Ours   (e) GT

Figure 5: Comparison with current methods at g2a direction on VIGOR-GEN

### 5.3 ABLATION STUDY

In this section, we conduct ablation studies to validate the effectiveness of each component in our method. We report variant models at the g2a direction on CVUSA. As the key design of our method, we first replace the retrieval embedder with a trainable pix2pix encoder (**i**). In this way, it is difficult for the model to transform the information from the source view to the target view, as there still exists a large domain gap.

The second experiment omits the attn-AdaIN in our model (**ii**). This modification loses the advantage of fusing retrieval embedding in the corresponding semantic region, which leads to a decrease in similarity.

Next, we also analyze the role of the style (**iii**) and retrieval embedding (**iv**) in our generator. The fusion of retrieved information and style improves the network from two perspectives: correspondence and diversity. First, by fusing embeddings in deep layers, the model ensures the generation of semantically consistent representations in the target view against the visual difference. We observe a degradation of the performance in various metrics from Table 4, especially in R@1 (with 9% drop). Sec-

Table 4: Ablation studies of our network on the CVUSA dataset.

| Method | CVUSA | | | | |
|---|---|---|---|---|---|
| | SSIM↑ | PSNR↑ | LPIPS↓ | FID↓ | R@1↑ |
| Ours | 0.3702 | 14.33 | **0.4302** | **13.57** | **96.25** |
| (**i**)w/o Embedder | 0.3312 | 13.66 | 0.4656 | 38.81 | 12.67 |
| (**ii**)w/o Attn-AdaIN | 0.3629 | 14.01 | 0.4461 | 16.51 | 89.42 |
| (**iii**)w/o Style | **0.3720** | **14.28** | 0.4412 | 17.88 | 94.23 |
| (**iv**)w/o Ret. | 0.3571 | 13.75 | 0.4377 | 15.74 | 87.67 |
| (**v**)Same Structure | 0.3490 | 14.06 | 0.4332 | 14.29 | 96.12 |
| (**vi**)w/o coarse D | 0.3454 | 14.11 | 0.4308 | 13.67 | 95.61 |

ond, the additional style information promotes the diversity of visual features and enriches the visual representations, which facilitates the generation of exclusive information in the target view. It has a slight increase in SSIM, but a significant drop in LPIPS and FID. We then analyze the role of different structures. If the model uses the same structure (i.e., ResBlock-S) to generate structural and facade information, metrics such as FID and LPIPS rise. Besides, the performance of the model degrades if the discriminator for coarse images is disabled. Consequently, facade generation modules reinforce the performance of the network in cross-view synthesis. To gain more insight into the attn-AdaIN, we visualize the mask $M$ learned on different feature levels in Figure 6, where the brighter pixel indicates the higher weight for retrieval embedding.

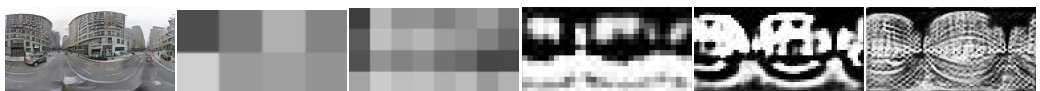

Figure 6: Visualization of the weight map $M$ on VIGOR-GEN

## 5.4 FURTHER DISCUSSION

The retrieval embedder bridges the domain gap and provides a stable direction of gradient descent. The embedder trained using retrieval loss is smooth in the embedding space. Once the model generates an incorrect identity of the target image, the embedding using retrieval loss can provide a good gradient

Table 5: The comparison of different Embedder in our generator on CVUSA at a2g.

| Embedder | CVUSA | | | | |
|---|---|---|---|---|---|
| | SSIM↑ | PSNR↑ | LPIPS↓ | FID↓ | R@1↑ |
| SAIG | **0.3706** | **14.32** | **0.4302** | **13.57** | **96.25** |
| LPN | 0.3559 | 13.89 | 0.4544 | 25.29 | 30.45 |

direction for the generator to change the identity correctly. In another type of embedder, which is trained on a discriminative task, the space can become non-smooth. Therefore, we compare the use of LPN Wang et al. (2021) in the generator, which regards cross-view image retrieval as a classification and thus applies instance loss Zheng et al. (2020). As shown in Table 5, the performance of the generator using LPN Wang et al. (2021) is significantly worse than the generator using SAIG.

## 6 CONCLUSION

In this work, we introduce a novel method for cross-view photo-realistic image synthesis. Specifically, we adopt a retrieval-guided framework that employs a retrieval network as the embedder and thus extracts information corresponding to the target view from the source images. Furthermore, we propose new generators for better-generating structure and facade, which facilitates correspondence and the generation of view-specific semantics in the target view. In addition, we also build a large-scale, more practical, and challenging dataset (VIGOR-GEN) in the urban setting. Through extensive experiments, it is verified that our method outperforms other competitive methods.

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

## A APPENDIX

### A.1 ARCHITECTURE

**One-way Discriminator**  The one-way discriminator is primarily employed in the text-to-image generation to guide the generation of images with consistent semantics as the text. There exists a domain gap between the two modalities describing the same content, similar to cross-view image pairs. A simple analogy is that the images in both views depict the same location, while the two images are greatly different in resolution and representation. We introduce this discriminator in the cross-view image synthesis task for its guidance of the corresponding content. The embeddings from the source view are spatially expanded and concatenated with the image features. For mismatched embedding-image pairs or non-realistic images, the discriminator will treat them as incorrect content. Tables 6 and 7 show the details of the one-way discriminator.

Table 6: Discriminator.

| Discriminator |
| --- |
| Synthesized image **(3, H, W)** |
| 4×4 Conv + LeakyReLU **(48, H/2, W/2)** |
| 4×4 Conv + LeakyReLU **(96, H/4, W/4)** |
| 4×4 Conv + LeakyReLU **(192, H/8, W/8)** |
| 4×4 Conv + LeakyReLU **(384, H/16, W/16)** |
| *concat* (embedding) **(768, H/16, W/16)** |
| 4×4 Conv + LeakyReLU **(96, H/32, W/32)** |
| 4×4 Conv + LeakyReLU **(1, H/64, W/64)** |

Table 7: Discriminator for coarse images.

| Discriminator |
| --- |
| Synthesized image **(3, H, W)** |
| 4×4 Conv + LeakyReLU **(48, H/2, W/2)** |
| 3×3 Conv + LeakyReLU **(96, H/2, W/2)** |
| 3×3 Conv + LeakyReLU **(192, H/2, W/2)** |
| 4×4 Conv + LeakyReLU **(384, H/4, W/4)** |
| *concat* (embedding) **(768, H/4, W/4)** |
| 4×4 Conv + LeakyReLU **(96, H/8, W/8)** |
| 4×4 Conv + LeakyReLU **(1, H/16, W/16)** |

Table 8: Generator.

| Generator |
| --- |
| Source Embedding **(384)** |
| Linear + reshape **(384, H/128, W/128)** |
| ResBlock-S **(384, H/64, W/64)** |
| ResBlock-S **(384, H/32, W/32)** |
| ResBlock-S **(384, H/16, W/16)** |
| ResBlock-S **(384, H/8, W/8)** |
| ⟶3×3 Conv + Tanh **(3, H/16, W/16)** |
| *concat* (Noise) **(512, H/8, W/8)** |
| ResBlock-T **(256, H/4, W/4)** |
| ResBlock-T **(128, H/2, W/2)** |
| ResBlock-T **(64, H, W)** |
| ⟶3×3 Conv + Tanh **(3, H, W)** |

**Generator**  The source embedding is fed into the generator and then concatenated a noise to recover the target view image. Each ResBlock upsamples the feature map. The detail of the generator is shown in Table 8.

**Retrieval Embedder**  We apply the Simple Attention-based Image Geo-localization backbone (SAIG) Zhu et al. (2023) as the embedder in our network. The SAIG backbone is a retrieval network for cross-view image geo-localization, which has two branches for encoding the ground-view images and the aerial-view images, respectively. We use a single branch with a fixed weight to embed the corresponding view images. Since the SAIG brings the image pairs closer in the embeddings space without any preprocessing, the embeddings can be regarded as the representation without a domain gap and thus support the generation. In this study, we utilize the variant model SAIG-S+GAP+ASAM+Triplet Loss to gain the source embeddings.

## A.2  VIGOR-GEN DATASET

The VIGOR dataset is originally proposed by Zhu et al. (2021) for the task of one-to-many cross-view image geo-localization, covering four cities: New York, Chicago, Seattle, and San Francisco. To fit the realistic scenarios, VIGOR is set up as a non-centrally aligned ground-aerial image of urban areas. Each ground-view panorama corresponds to one positive aerial image and three semi-positive images for the retrieval task. Inspired by VIGOR, to alleviate the shortage of datasets for cross-view image synthesis in urban areas, we build a derived dataset VIGOR-GEN from VIGOR. Moreover, to improve the stability and quality of the dataset, we remove the meaningless image pairs, including those located in water, indoors, or with a lot of mosaic and distortion, etc. The VIGOR-GEN dataset will be publicly available.

A.3 ADDITIONAL QUANTITATIVE RESULTS

Following previous work Regmi & Borji (2018); Lu et al. (2020); Toker et al. (2021); Shi et al. (2022), we adopt the widely used *Structural-Similarity (SSIM)*, *Peak Signal-to-Noise Ratio (PSNR)* and *Learned Perceptual Image Patch Similarity (LPIPS)* Zhang et al. (2018) to measure the similarity between the synthesized and real images at the pixel-wise level and feature-wise level, respectively. Meanwhile, the realism of the generated images is measured by *Fréchet Inception Distance (FID)* Heusel et al. (2017), which measures the feature distribution by a pre-trained Inception v3 network. We extract retrieval embeddings using another cross-view image retrieval model SAIG-D Zhu et al. (2023) and measure the recall accuracy from generated images to source images. We report the Recall@1 (R@1) in our experiment, which indicates whether the first image returned by the retrieval network is correct. A higher R@1 implies the generated images preserve the identity information better and show a higher correspondence with the target-view image at the feature-wise level in terms of retrieval. The performance of all compared methods was measured using source code replication. Some methods use earlier versions of the code[1] to measure the performance, which leads to significant differences in SSIM. For a fair comparison, after we acquired the generated images, we chose to measure **SSIM** and **PSNR** by the code of S2SP Shi et al. (2022)[2] instead of the code[3]. **LPIPS** is calculated by this code[4]. **FID** is calculated by this code[5], whereas the reference images are the validation set of the corresponding dataset.

**Training Details** We train our model with 200 epochs using Adam Kingma & Ba (2014) optimizer and $\beta_1 = 0.5, \beta_2 = 0.999$. The learning rate of the generator and discriminator is set to 0.0001 and 0.0004, respectively. For each dataset, we use the maximum possible batch size on 4 32GB NVIDIA Tesla V100 GPUs (bs=32 for CVUSA, bs=24 for CVACT, and bs=24 for VIGOR-GEN). The diversity loss is computed every 4 steps. We use DiffAug Zhao et al. (2020) {Color, Cutout} as a data augmentation strategy during the training. The $\lambda_{rec}$, $\lambda_{perc}$ and $\lambda_{id}$ is set to 50, 50 and 10. The $\lambda_{div}$ is set to 0.1 in CVUSA and CVACT, while is set to 1 in VIGOR-GEN. In previous work, the val set was considered as the test set, note that we only took the final checkpoint for testing and did not select the intermediate checkpoints.

A.4 ADDITIONAL QUANTITATIVE RESULTS

**More Realistic** We present more synthesized images to demonstrate the effectiveness of our method in Figure 7, 4, 8, 9. Compared to other methods, our model generates images with clearer roads, which are more realistic. This demonstrates that our model performs well in generating exclusive information as well as resolving visual differences.

**Higher Quality** To better demonstrate the capability of our model, we perform higher resolution ($256 \times 1024$) cross-view image synthesis. Compared to the existing methods (SelectionGAN Tang et al. (2019) and PanoGAN Wu et al. (2022)), our method performs better in all metrics, shown in Table 9.

A.5 FURTHER DISCUSSION

**Embedder** The retrieval embedder not only compensates for the domain gap problem in cross-view image synthesis but also provides a stable gradient descent direction, making the generator easier to train.

The embedder trained using retrieval loss is smooth in the embedding space. Once the model generates an incorrect identity of the target image, the embedding using retrieval loss can provide a good gradient direction for the generator to correctly change the identity. However, in a non-smooth

---

[1] `https://github.com/kregmi/cross-view-image-synthesis/blob/master/Evaluation/`

[2] `https://github.com/YujiaoShi/Sat2StrPanoramaSynthesis/tree/main/evaluation_metrics`

[3] `https://github.com/kregmi/cross-view-image-synthesis/blob/master/Evaluation/compute_ssim_psnr_sharpness.lua`

[4] `https://github.com/richzhang/PerceptualSimilarity`

[5] `https://github.com/mseitzer/pytorch-fid`

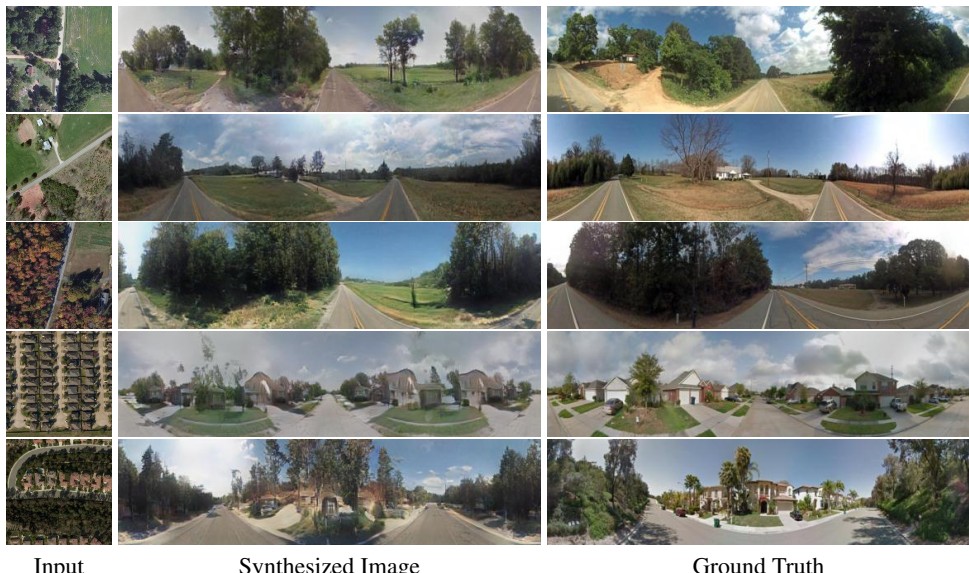

Input                    Synthesized Image                    Ground Truth

Figure 7: Our synthesized images on CVUSA

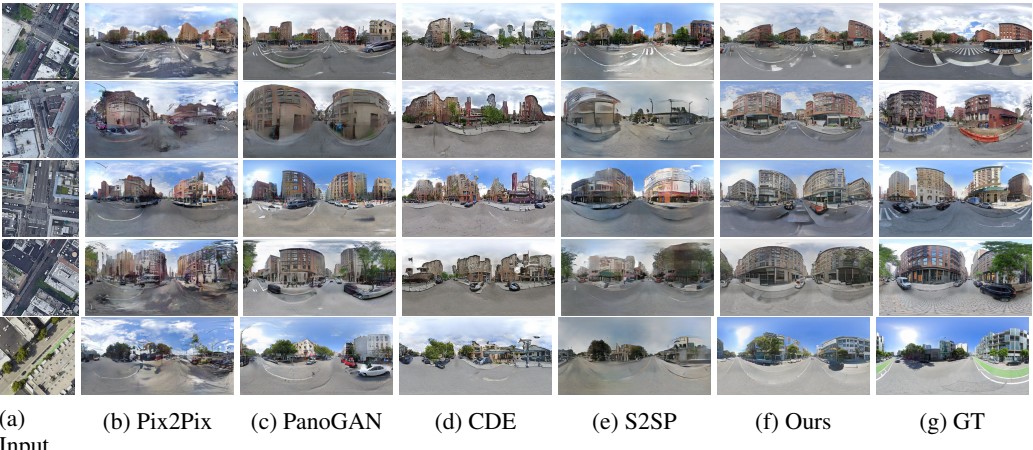

(a)        (b) Pix2Pix    (c) PanoGAN    (d) CDE    (e) S2SP    (f) Ours    (g) GT
Input

Figure 8: Comparison with current methods Pix2Pix Isola et al. (2017), PanoGAN Wu et al. (2022), CDE Toker et al. (2021) and S2SP Shi et al. (2022) on VIGOR-GEN.

embedding space, the embedding would make discrete jumps that block the identity from being corrected.

Table 9: The comparison of different methods with our generator on CVUSA at $256 \times 1024$.

| Dataset | Method | SSIM↑ | PSNR↑ | LPIPS↓ | FID↓ | R@1↑ |
|---------|--------|-------|-------|--------|------|------|
| CVUSA | SelectionGAN | 0.4010 | 13.21 | 0.6169 | 103.27 | 3.81 |
| | PanoGAN | 0.3575 | 13.47 | 0.5566 | 81.91 | 30.58 |
| | Ours | **0.4232** | **14.11** | **0.4978** | **17.88** | **96.01** |
| CVACT | SelectionGAN | 0.4876 | 14.28 | 0.5232 | 97.63 | 5.76 |
| | PanoGAN | 0.4915 | 14.31 | 0.4959 | 86.61 | 23.02 |
| | Ours | **0.5513** | **14.48** | **0.4938** | **24.62** | **86.91** |
| VIGOR-GEN | SelectionGAN | 0.4154 | 13.11 | 0.5225 | 106.24 | 7.80 |
| | PanoGAN | 0.4229 | 13.68 | 0.4933 | 79.72 | 8.26 |
| | Ours | **0.4771** | **14.01** | **0.4876** | **23.54** | **36.18** |

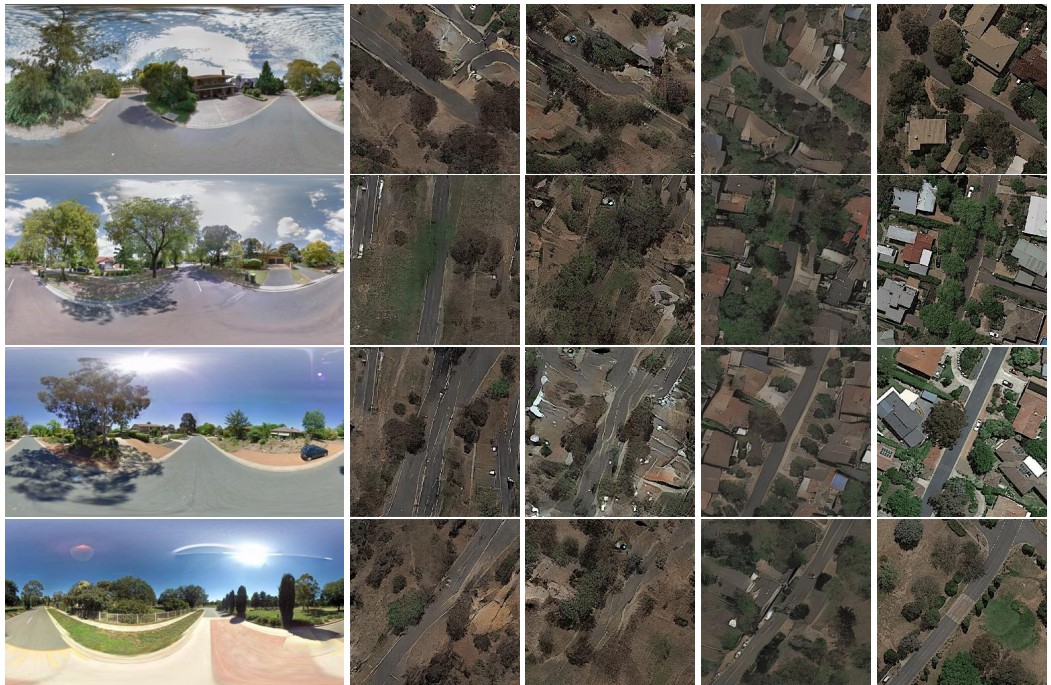

Figure 9: Comparison with current methods at g2a direction on CVACT

Moreover, we compare the use of LPN Wang et al. (2021) in the generator, which regards a cross-view image retrieval as a classification and thus applies the instance loss Zheng et al. (2020). This type of embedder, which is trained on a discriminative task, makes the space non-smooth. Although the classification embedder can also bridge the domain gap, its generation performance and convergence speed are significantly lower than that of the generator using the retrieval embedder, as shown in Table 5 and Figure 10.

We believe that the key to cross-view image synthesis lies in not only how to design the models that bridge the domain gap, but also how to integrate them organically with the generative model. We also believe this finding will shed more light on future cross-view image synthesis.

To quantify the smoothness of different cross-view models (e.g., SAIG and LPN), we perform a visual analysis of them. We first randomly pick up two aerial-view images from the test set and then compute the interpolations in the embedding space. For each of the interpolating points, we retrieve the closest images from the train set. Commonly, if the embedding space is smooth, the embedder is going to exhibit continuously changing identities Kim et al. (2022), whereas others show repeated identities, imply-

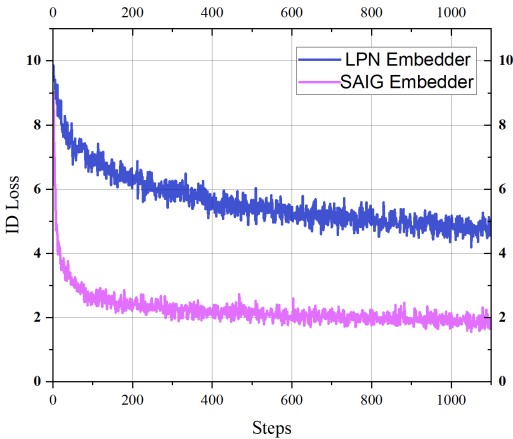

Figure 10: The curve of ID loss in generator using different embedder.

ing non-smoothness. As shown in Figure 11, the images retrieved by the interpolated embedding of the Retrieval embedder are smoother in terms of identities. The first row and the last row (the twelfth row) are two randomly selected images from the test set, the interpolated result between two embeddings is retrieved in the training set and the results are shown from the second row to the eleventh row. It can be seen that the images retrieved using SAIG embedder have continuously changing identities.

LPN Embedder  SAIG Embedder  LPN Embedder  SAIG Embedder

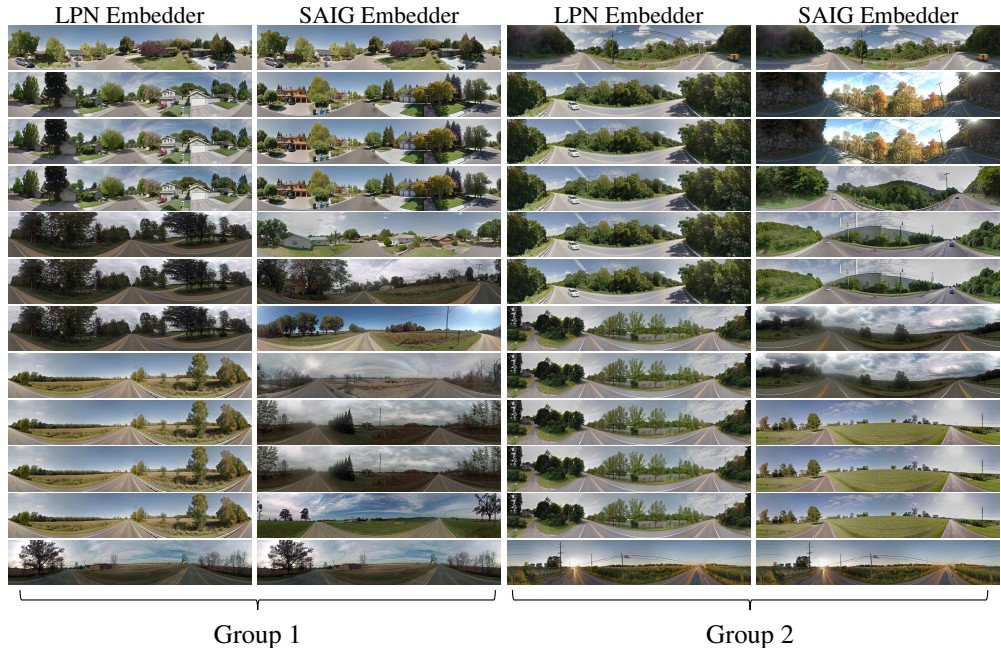

Group 1  Group 2

Figure 11: The retrieval results of different embedder.

**Different Residual Block**  In Figure 12, we show the effect of using different residual blocks in the experiments. It can be observed that the images generated using only Structure-S have more artifacts. This can be further illustrated by the ablation study in the main paper.

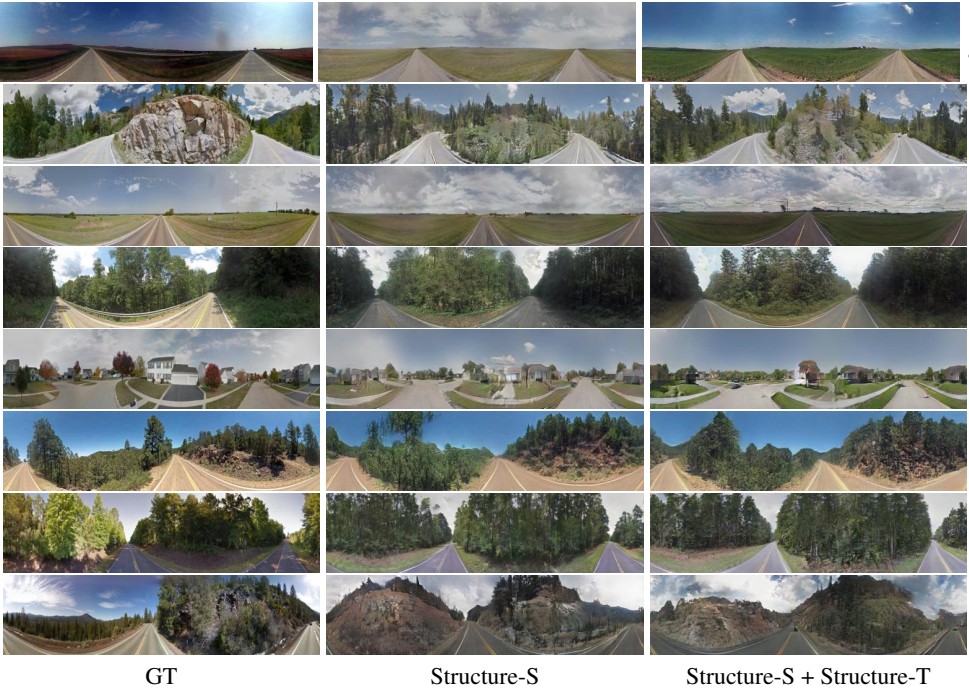

GT  Structure-S  Structure-S + Structure-T

Figure 12: Comparison of images generated by models using different Residual Blocks.

Table 10: The comparison of model size with different model.

| Model | #Params | FPS | FID |
|---|---|---|---|
| Pix2Pix | 41.8M | 34.1 | 82.84 |
| XFork | 39.2M | 33.8 | 79.75 |
| SelectionGAN | 58.3M | 18.0 | 90.72 |
| PanoGAN | 88.0M | 19.1 | 75.24 |
| CDE | 37.3M | 35.9 | 20.63 |
| S2SP | 33.6M | 22.1 | 44.15 |
| Ours | **25.9M** | **39.2** | **13.57** |

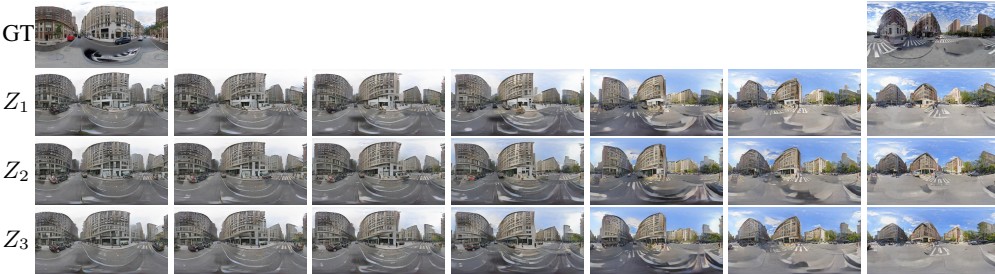

Figure 13: Generated images using embeddings interpolated from two random images and different $Z_{local}$. The middle image is the result generated from the intermediate latent code between two images. It can be observed that the generated images change smoothly with changing embeddings, as well as the facades change with different $Z_{local}$.

**Model Size**    To better illustrate the overhead of our model, we show the comparison with different models in terms of model size and speed. As illustrated in Table 10, our model has fewer parameters and faster inference compared with other methods.

**Interpolated Embedding**    Moreover, we show the effect of different $Z_{local}$ on generating exclusive information in Figure 13. Different $Z_{local}$ render different building facades for images of the same structure. We also randomly pick two embeddings and interpolate them as the input, where a smooth change in identity can be observed.

