# OpenReview forum: "Retrieval-guided Cross-view Image Synthesis"
_ICLR.cc/2025/Conference — ICLR 2025 Conference Withdrawn Submission_

### Official Review · Reviewer_SF5W · 2024-10-31

**Soundness:** 2
**Presentation:** 2
**Contribution:** 2
**Rating:** 3
**Confidence:** 4

**Summary:**

- The work proposes a novel technique to perform cross-view image generation using satellite and ground-view image pairs leveraging generative modeling.
- The authors also introduce a cross-view dataset for urban scenes by extending VIGOR.
- Notable performance improvements were reported in comparison to selected baselines.

**Strengths:**

- The technique uses existing methods for mapping, retrieval and GANs to composing an effective solution towards cross-view synthesis.
- The dataset introduced fills a gap by providing urban setting data.
- Improved correspondence was seen between cross-view pairs relative to reported methods.

**Weaknesses:**

- W1: Motivation of the work is not well-grounded. The manuscript lists out application areas (L48) without specifying in sufficient detail how cross-view image synthesis as presented in the current work would fit in those applications. Another way to further stengthen this weakness would be to demonstrate how this work would be used down-stream empirically. For instance, if this is meant to be useful for cross-view localization which also uses common datasets employed in this work, then down-stream results with and without this work discussed here can be shown.
- W2: The closest area I’ve seen where cross-view synthesis between satellite and ground-view image pairs have been beneficial is in performing cross-view retrieval or finer localization for autonomous driving. However, in such applications, including AR/VR or remote-synthesis, genrating true representation of the physical world is necessary and not just scenes with high visual realism. If not, this would raise serious safety/ethical issues. By design, this is an ill-posed problem and generative models in the way they’re used here do not offer the right trade-off between factuality and creativity for the application setting of this work. For instance, in Figures 5 (satellite generation), 7 and 8 (ground-view generations) there are quite a few scene elements that do not exist in the physical world but exists in the synthetic generations which can mislead localization results or end-users. Therefore, I am not convinced this work solves the listed problems. It is imperative to minimize hallucinations of generative models to be useful in such problems, one example of doing this is prior work [1] listed below.
- W3: The manuscript is missing representative works which have same goals as this work. They also seem to operate in urban settings as targeted in this work.
    - [1] Geospecific View Generation - Geometry-Context Aware High-resolution Ground View Inference from Satellite Views (ECCV 2024) - https://arxiv.org/pdf/2407.08061
    - [2]: Sat2Scene: 3D Urban Scene Generation from Satellite Images with Diffusion - https://arxiv.org/pdf/2401.10786
- W4: Evaluation: I also think evaluation metrics like PSNR will be heavily influenced by the sky regions. They occupy most of the image and therefore these metrics might reflect the reconstruction quality of sky and not necessarily the effectiveness of the methods.
- W5: Writing
    - The manuscript states one of the advantages of the method is the lack of reliance on segmentation maps owing to computations and because it complicates the reverse generation process. This is unclear - this setting is meant to be an offline setting, so I doubt computation is a decisive factor especially if it provides benefits in bridging domain gap. It is not clear to me how segmentation maps complicate reverse generation, the statements need to be more specific and targeted.
    - L122: It is claimed referenced prior work has a complicated pipeline without describing how.
    - L137: It is stated it is essential to study a GAN model before moving towards diffusion model - Why is this true? Is there something about this particular problem/context that gives credence to this statement?
    - Figure 3 needs significantly more support to be reader friendly. It is very difficult to understand which is the structure generator, the facade generator by looking at the figure without having to go back and forth multiple times from text to figure.
- Minor:
    - Wrong paper seems to have been cited for CrossViewDiff.
    - Sec 2: Author names are repeated twice throughout citations.
    - L222: sigma is standard deviation not variance.
    - Fig 9 in appendix does not have method names.
    - Typo in L349

Weighting weaknesses as W1 = W2 > W3 > W4 > W5

**Questions:**

- Q1: Have the authors considered using geometry information from the cross-view pairs to condition the generations? I would expect the scenes to be more constrained and closer to the real-world and therefore be more useful for the applications mentioned. [1] also shows this to be the case.
- Q2: What was the motivation behind using GANs over diffusion models? Was it empirically based or just preference?
- Q3: L381: It is unclear to me why semantic maps were discarded from SelectionGAN for a fair comparison (Tab 3). If anything, this makes it unfair if I understand correctly since an important source of information pivotal to the baseline method is being removed. Am I misunderstanding?

---

### Official Review · Reviewer_3zKp · 2024-11-01

**Soundness:** 3
**Presentation:** 2
**Contribution:** 3
**Rating:** 5
**Confidence:** 5

**Summary:**

This paper proposed a retrieval-guided cross-view synthesis framework, which leverages retrieval features to bridge the domain gap between ground and satellite images. This paper further introduces a new generator that tries to enhance the semantic consistency and diversity of the generated images by the retrieval features and random noises. Additionally, this paper presents a new dataset, VIGOR-GEN, for cross-view synthesis benchmarking. Experimental results on three standard benchmarks, including CVUSA, CVACT and VIGOR-GEN, have demonstrated the effectiveness of the proposed method.

**Strengths:**

+ Leveraging viewpoint invariant retrieval features from cross-view synthesis is interesting.

+ The introduction of random noise to enhance the diversity makes sense, and the satellite and ground cross-view image synthesis is inherently a one-to-many task due to severe occlusions and different illumination/weather conditions.

+ The proposed method achieves state-of-the-art performance over three benchmarks.

**Weaknesses:**

1. While this paper achieves state-of-the-art performance, I am concerned about the technical contributions' novelty and soundness.

(a). My overall understanding of this paper's contribution is that this paper leverages retrieval features for cross-view synthesis, which is nice but somewhat incremental. Other modifications, such as attentional AdaIN block and noise injection, are not new in the GAN community. Furthermore, it seems this paper has incorporated many engineering network architecture designs, such as first generating low-resolution images and then making a high-resolution one. It is not clear whether these designs makes the performance good or others.

(b). Injecting noise to enhance diversity is interesting. However, other than reconstruction quality with respect to ground truth data, this paper does not show any qualitative evaluation of the diversity of the generated images. Will different noises lead to different images generated based on the same source image? What are the differences? Following this question, the diffusion model has shown superior performance to GAN in many tasks. What is the superiority of the proposed method over diffusion models?

(c). As the introduction claims, how does the proposed generator incorporate view-specific semantics, and why? How to enhance semantic consistency (and diversity)?

2. This paper argues that previous works leverage semantic maps or reprocessing modules, which increases the computation burden. However, the model size and computation complexity (training and inference speed) are not compared. The proposed method seems to have a larger complexity than previous works from the network architecture perspective. How long does the training take, what kind of GPUs are used?

3. For the newly proposed VIGOR-GEN, why has the number of ground images changed? What kind of ground images are ignored?

4. I appreciate the authors’ effort in collecting center-aligned satellite images. However, I have a question about the original VIGOR dataset, which includes GPS labels for both ground and satellite images. Could the satellite images not be shifted based on these GPS labels to achieve center alignment? While the authors might argue that this approach could compromise the resolution or coverage of the satellite images, I believe that coverage is less critical for satellite-to-ground synthesis. In urban areas, the visible distance of a ground image is quite limited, especially given that the original satellite images cover approximately 75m x 75m. The ground camera's location falls within about 18m of the satellite image center, allowing for at least a 36m x 36m satellite patch to be derived that is center-aligned with a ground panorama, which corresponds to roughly 18m of visible distance. Isn’t this sufficient? Is there any comparison of the different coverages of satellite images in the cross-view synthesis task?

**Questions:**

Plz refer to my comments in "weakness".

---

### Official Review · Reviewer_Czjr · 2024-11-03

**Soundness:** 3
**Presentation:** 3
**Contribution:** 3
**Rating:** 5
**Confidence:** 5

**Summary:**

For the cross-view image generation task, this paper constructs a derivative dataset, VIGOR-GEN, based on the existing VIGOR dataset and proposes an innovative approach to enhance the quality and consistency of cross-view image generation. This method incorporates a retrieval network during the encoding phase to maximize the preservation of structural information and semantic features from the input image, thereby guiding the generator model to achieve high-quality translation from satellite images to street-view images. This approach not only provides new data resources for cross-view generation tasks but also introduces a novel technical perspective for research in this field.

**Strengths:**

This paper proposes a retrieval-guided framework for cross-view image synthesis, which utilizes a retrieval network as an embedder to effectively address the domain gap between different views. This approach preserves both shared and view-specific semantic information while optimizing the generation process, thereby enhancing the quality and practical utility of cross-view image synthesis.
The newly introduced VIGOR-GEN dataset enriches urban cross-view image synthesis, offering realistic center-aligned street-satellite pair.

**Weaknesses:**

Regarding quantitative evaluation, the a2g task is compared with diffusion-based methods and achieves promising results. However, it appears that the g2a task is only compared with GAN-based methods. Could this imply that the proposed approach may have some limitations in g2a performance relative to diffusion-based models? To the best of my knowledge, there appear to be the following similar works: AerialDiffusion[1],SkyDiffusion[3], and Cross-View Meets Diffusion[2] .

[1] Kothandaraman D, Zhou T, Lin M, et al. Aerial Diffusion: Text Guided Ground-to-Aerial View Translation from a Single Image using Diffusion Models[J]. arXiv preprint arXiv:2303.11444, 2023.
[2] Arrabi A, Zhang X, Sultan W, et al. Cross-View Meets Diffusion: Aerial Image Synthesis with Geometry and Text Guidance[J]. arXiv preprint arXiv:2408.04224, 2024.
[3] Ye J, He J, Li W, et al. SkyDiffusion: Street-to-Satellite Image Synthesis with Diffusion Models and BEV Paradigm[J]. arXiv preprint arXiv:2408.01812, 2024.

**Questions:**

In the urban scene experiments, the paper uses the VIGOR-GEN dataset, rather than the VIGOR dataset, for comparisons. According to the description, the VIGOR-GEN dataset contains image pairs of street views and satellite images aligned at the center. Could this reveal that the algorithm might encounter challenges in effectively utilizing satellite image information from datasets that are not center-aligned during the generation stage?

---

### Official Review · Reviewer_nz3Y · 2024-11-03

**Soundness:** 2
**Presentation:** 1
**Contribution:** 2
**Rating:** 3
**Confidence:** 4

**Summary:**

This paper proposes a Retrieval-Guided Framework for cross-view image synthesis and extends the VIGOR dataset to create VIGOR-GEN, incorporating satellite images collected from the Google Maps API.

**Strengths:**

1. This paper achieves cross-view image synthesis without relying on additional semantic segmentation maps.
2. The proposed method demonstrates state-of-the-art performance.
3. The ablation studies are comprehensive.

**Weaknesses:**

1. Figure 3 contains question marks (mojibake), making it difficult to interpret. Additionally, the textual description of the overall architecture lacks clarity, which hinders understanding of the details.
2. The motivation for not using preprocessing is unconvincing. The authors claim that polar transformation or geographic projection is computationally burdensome and complex, but these methods are not computationally intensive compared to a network. Additionally, if the decision to avoid these methods is to produce more realistic images, the authors fail to explain why these methods would lead to less realistic results.
3. The dataset contribution primarily involves incorporating data downloaded from the Google Maps API, which is subject to Google’s usage restrictions and therefore cannot be considered a contribution.

**Questions:**

This method heavily relies on the training dataset. Does it perform well on entirely unseen scenes?

---

### Official Review · Reviewer_NQLS · 2024-11-04

**Soundness:** 3
**Presentation:** 3
**Contribution:** 2
**Rating:** 5
**Confidence:** 4

**Summary:**

This paper introduces a retrieval-guided framework for cross-view image synthesis. Specifically, the proposed method employs a retrieval model as an embedder to identify view-invariant semantics within a specific view, ensuring consistency of these semantics in the generated images across both views. Experimental results demonstrate that this approach outperforms previous state-of-the-art methods in terms of performance. However, in my opinion, this work is akin to an incremental advancement, merely substituting one embedder for another, which limits its overall innovativeness.

**Strengths:**

- The proposed method demonstrates superior performance compared to previous approaches, and the overall writing is commendable.
- The paper structure is well organized and easy to understand.

**Weaknesses:**

- This work is akin to an incremental advancement, merely substituting one embedder for another, which limits its overall innovativeness.
- Could you provide a visual comparison with CROSSVIEWDIFF [1]?
- Could you present some visual examples to illustrate better the advantages of selecting a retrieval model as an embedder?
- There are some citations of methods in the paper that need correction, such as CROSSVIEWDIFF, which should be amended.

[1] CROSSVIEWDIFF: A CROSS-VIEW DIFFUSION MODEL FOR SATELLITE-TO-STREET VIEW SYNTHESIS

**Questions:**

See Weaknesses

---

### Note · Authors · 2024-11-28

I have read and agree with the venue's withdrawal policy on behalf of myself and my co-authors.